# Proteomic Analysis Revealed the Antagonistic Effect of Decapitation and Strigolactones on the Tillering Control in Rice

**DOI:** 10.3390/plants13010091

**Published:** 2023-12-27

**Authors:** Yanhui Zhao, Manrong Zha, Congshan Xu, Fangxu Hou, Yan Wang

**Affiliations:** 1College of Biology Resources and Environmental Sciences, Jishou University, Jishou 416000, China; huibo02020219@163.com (Y.Z.); zmr0729@163.com (M.Z.); accobear@163.com (F.H.); 2Key Laboratory of Plant Resources Conservation and Utilization, College of Hunan Province, Jishou 416000, China; 3Anhui Science and Technology Achievement Transformation Promotion Center, Anhui Provincial Institute of Science and Technology, Hefei 230002, China; xucongshan1027@163.com

**Keywords:** rice, tiller bud, decapitation, strigolatones, sucrose metabolism

## Abstract

Removing the panicle encourages the growth of buds on the elongated node by getting rid of apical dominance. Strigolactones (SLs) are plant hormones that suppress tillering in rice. The present study employed panicle removal (RP) and external application of synthesized strigolactones (GR) to modulate rice bud growth at node 2. We focused on the full-heading stage to investigate proteomic changes related to bud germination (RP-Co) and suppression (GR-RP). A total of 434 represented differentially abundant proteins (DAPs) were detected, with 272 DAPs explicitly specified in the bud germination process, 106 in the bud suppression process, and 28 in both. DAPs in the germination process were most associated with protein processing in the endoplasmic reticulum and ribosome biogenesis. DAPs were most associated with metabolic pathways and glycolysis/gluconeogenesis in the bud suppression process. Sucrose content and two enzymes of sucrose degradation in buds were also determined. Comparisons of DAPs between the two reversed processes revealed that sucrose metabolism might be a key to modulating rice bud growth. Moreover, sucrose or its metabolites should be a signal downstream of the SLs signal transduction that modulates rice bud outgrowth. Contemplating the result so far, it is possible to open new vistas of research to reveal the interaction between SLs and sucrose signaling in the control of tillering in rice.

## 1. Introduction

Tillering of rice (*Oryza sativa* L.) is a highly controlled agronomic trait that contributes to yield. Tillers grow from the tiller bud, located on the distal portion of the tiller base or the elongated upper internodes. Rice tillering occurs in two stages: the formation of an axillary bud at each leaf axil, followed by its outgrowth [1]. Tiller development is dynamic and adjustable under the impact of various factors. The primary shoot and inflorescence are considered to be major. For a bud located at the second node from the panicle, panicle removal can stimulate it to grow effectively [2,3,4].

Two exclusive models, the “transport canalization” and the “second messenger” models, were proposed to explain the apical dominance. In the auxin transport canalization model, bud outgrowth requires auxin export. Auxin flow in the main stem can inhibit axillary bud outgrowth by preventing the establishment of their own polar auxin transport (PAT) for auxin efflux from axillary buds [5,6]. In the second messenger model, cytokinins (CKs) and strigolactones (SLs) were regarded as the downstream messengers of auxin that regulate outgrowth upon entering axillary buds [7,8]. CK is the only phytohormone that stimulates bud outgrowth, and SLs function antagonistically by promoting CK degradation in buds [9,10,11,12]. Both SLs and CKs affect bud outgrowth by regulating *BRANCHED1* (*BRC1*) expression in buds, which is a major regulatory nexus for shoot branching. However, a few phenomena cannot be explained well by these two models [8,13,14,15]. Reduction of auxin content in stems by decapsulation, stem girdling, or application of auxin transport inhibitors does not always promote axillary bud germination. When the stem tip and the axillary bud phase are far apart, the axillary bud growth after decapitation occurs before the change in auxin content in the stem. In addition, SLs have also been found to prevent auxin transport through reduced PIN-FORMED1 (PIN1) accumulation in xylem parenchyma cell basal membranes, which act as the auxin facilitators that can establish the endogenous auxin gradient [16].

Currently, sugar availability is proposed as a significant modulator of shoot branching that alleviates the negative effect of strigolactones in many species [17,18,19,20,21,22,23,24,25,26]. The apical meristem curtails the outgrowth ability of buds by inhibiting the import of sugar into them [18]. Dormant buds are under a sugar starvation condition and can be stimulated by an increased sugar supply to whole plants or stem explants [21,27]. After decapitation, the dormant axillary buds start to grow, accompanied by the sugar level increase in buds [18]. Other studies using mutants have shown the same results, which revealed sugars could be the limiting factor for bud outgrowth [17,23,24,25]. In the absence of external carbohydrates, the nucleus would be compacted, thus preventing cell cycle progression, and could be one of the factors hindering bud growth [28]. At the same time, sugar can also act as a signal that promotes bud outgrowth [19]. Glycolysis, the tricarboxylic acid (TCA) cycle, and the oxidative pentose phosphate pathway (OPPP) were all proved involved in the sugar-dependent promotion of bud growth [29,30]. Sucrose is the main long-distance transported sugar in vascular plants [26,31] and is proposed to have a signaling function in bud growth regulation [20,29]. Since sugars have the opposite effect to SLs in branching regulation, interactions between sugars and SLs have been discussed recently [29,32]. Sucrose-stimulated bud outgrowth was proved to be preceded by downregulated SLs signaling gene expression [33].

This study aimed to identify underlying molecular processes affiliated with the antagonistic function of decapitation and endogenous SLs on bud growth regulation. We established three types of buds to simulate two different processes: decapitation-induced bud activation and SL-induced bud dormancy. The iTRAQ approach was the profile of differentially abundant proteins (DAPs) within three treatments to determine the significant processes related to the decapitation and SLs on bud growth regulation.

## 2. Materials and Methods

### 2.1. Plant Growth

An Indica cultivar, Yangdao 6, was used in this study, and rice plants were grown in a net house at Nanjing Agriculture University (Nanjing, China) during the rice-growing season. Twenty days after seeding, rice plants were transferred to a plastic pot (30 × 30 cm^2^) containing 15 kg of sieved sandy loam soil from the experimental field. Three days prior to transplanting the seedlings, 1.6 g of urea, 0.8 g of single superphosphate, and 1.2 g of KCl were applied per pot by mixed with soil. A rice water layer was applied to irrigate these pots over the entire growing season of the plants.

### 2.2. Treatments

After the full-heading stage, plants were divided into three groups, viz. normal growth conditions were continued for one group of plants as a control (Co); the second group with removed panicle (RP). For the third group, panicle removal was followed by applying 2 μM of rac-GR24 (a synthetic SLs) directly on the bud located at the second node from the panicle (node 2) by using a syringe. As leaf sheaths always cover the buds on node 2, peel out the sheaths from the stem to inject the solution into the gap between the stem and sheaths. We also supplied the same solution without GR24 to the Co and RP treatments to achieve uniformity among treatments in all other aspects. All the buds on node 2 were sampled with a snap-frozen in liquid nitrogen and stored at −80 °C for further analysis.

### 2.3. iTRAQ Protein Identification and Bioinformatics Analysis

Proteins were extracted from the tiller bud on node 2 from the panicle by using the trichloroacetic acid (TCA)/acetone precipitation methods [31]. To prepare the tiller buds for analysis, buds were first powdered using liquid nitrogen and then suspended in 35 mL of chilled acetone with 10% TCA. The mixture was then incubated at −20 °C for 2 h and subsequently centrifuged at 7830 rpm for 30 min at 4 °C. The supernatant was carefully removed, and the precipitation was washed three times with acetone. The resulting proteins were air-dried and dissolved in 600 μL of SDT buffer, which was composed of 4% SDS, 100 mM Tris-HCl, and 1 mM DTT, and had a pH of 7.6. The mixture was then boiled in water for 5 min, sonicated at 100 Watts, and boiled again for 5 min before being centrifuged at 13,400 rpm for 30 min at 4 °C. The supernatant was collected as a soluble protein fraction, and its concentration was measured using a Bradford protein assay kit with BSA as a standard. To ensure the quality of each protein sample, SDS-PAGE was used to test them. Additionally, Tricine-SDS-PAGE was used to verify the quantitative results from the Bradford assay and determine the quality of the extract.

The iTRAQ proteomic study was performed by Shanghai Boyuan Company Ltd. (Shanghai, China) Around 80 micrograms of peptides in each sample were labeled using iTRAQ reagents as per the manufacturer’s instructions (Applied Biosystems, Thermo Fisher Scientific Corporation, Waltham, MA, USA) by using iTRAQ 8-plex kits (AB Sciex Inc., Framingham, MA, USA). After labeling, the samples were combined and lyophilized. Next, we fractionated the iTRAQ-labeled peptides using strong cation exchange (SCX) chromatography in an AKTA Purifier 100 system from GE Healthcare, Chicago, IL, USA, equipped with a polysulfethyl column (PolyLC Inc., Columbia, MD, USA). We eluted the peptides at a flow rate of 1 mL/min, using two buffers—Buffer A (10 mM KH_2_PO_4_ and 25% *v/v* ACN, pH 3.0) and Buffer B (10 mM KH_2_PO_4_, 25% *v/v* ACN, and 500 mM KCl, pH 3.0). The two buffers were filter-sterilized. We used the following gradient for separation: 100% Buffer A for 25 min, 0–10% Buffer B for 7 min, 10–20% Buffer B for 10 min, 20–45% Buffer B for 5 min, 45–100% Buffer B for 5 min, 100% Buffer B for 8 min, and finally, 100% Buffer A for 15 min. We monitored the elution process by measuring absorbance at 214 nm and collected fractions every 1 min. We combined the collected fractions into eight pools and used Empore TM SPE Cartridges C18 (standard density), Sigma-Aldrich, St. Louis, MO, USA to desalt them. We concentrated each fraction via vacuum centrifugation and reconstituted them in 40 μL of 0.1% *v/v* trifluoroacetic acid. Finally, we stored all samples at −80 °C until LC-MS/MS analysis.

Differentially abundant proteins (DAPs) were identified according to a fold change >1.2 and FDR < 0.05. Functional analysis of the proteins identified was conducted using GO annotation (http://www.geneontology.org/ (accessed on 30 August 2018)), and proteins were categorized according to their biological process, molecular function, and cellular localization [34]. TBtools performed Kyoto Encyclopedia of Genes and Genomes (KEGG) pathway enrichment analysis of the DAPs (http://cj-chen.github.io/tbtools/ (accessed on 30 August 2018)) [35].

### 2.4. Detection of Sucrose Content in the Bud

To measure the sucrose content, a detection kit (Solarbio Life Sciences, Beijing, China) was used for the sucrose content detection and performed according to the manual. Over 60 buds at node 2 were collected for each sample, all the buds were ground at room temperature with 1 mL of kit extract. After that, an 80 °C water bath extraction was conducted for 10 min and then centrifuged for 10 min (4000× *g* and 25 °C). Carbon powder was introduced to the supernatant and left for 30 min to observe decolorization. Next, a 1 mL extract was added, and the mixture was centrifuged for 10 min (4000× *g* and 25 °C) until the final supernatant was obtained. The optical density of the supernatant was measured at 480 nm using a UV-Vis spectrophotometer (V-5100; METASH, Shanghai, China) through colorimetric analysis.

### 2.5. Enzyme Activities Analysis

Similar methods using enzyme reagent kits were employed to detect the activity of sucrose vacuolar invertase (vINV) and sucrose synthase (SuSy). A total of 1 mL of vINV and SuSy extracts were added to 0.2 g of buds (consisting of over 30 buds) that were collected from node 2. The samples were then ground in liquid nitrogen, homogenized in an ice bath, and low-temperature centrifuged for 10 min (8000 r/min and 4 °C). The resulting supernatant liquid was collected, and the related enzyme activity was determined according to the enzyme kit instructions. The final reaction products were determined at different λ (SuSy: 480 nm; vINV: 510 nm) by the enzyme using a UV-Vis spectrophotometer.

### 2.6. Statistical Calculations

Results were analyzed using SPSS software v16.0 for Windows. Data from each sampling were analyzed separately. Means were tested with Student’s *t*-test and the significance level was *p* < 0.05.

## 3. Results

### 3.1. Effects of Panicle Remove (RP) and rac-GR24 (GR) Treatment on Bud Growth in Rice

To determine the effectiveness of our treatments in controlling bud outgrowth in rice, we focused on the dormant bud on node 2 at the full-heading stage. Two treatments, “removed panicle (RP)” and “rac-GR24 (synthetic strigolactones) supply” after RP (GR), were performed to modulate the transition between dormancy and germination of bud. Bud length was measured within 72 h after treatments (Figure 1). Bud length on node 2 of intact plants (Co) remained stable, indicating the dormant condition that makes all the buds we observed comparable for further treatment. Dormant buds were activated after RP treatment, and the RP’s bud length was significantly longer than the Co after 12 h of treatment. Not surprisingly, GR treatment decelerated the bud outgrowth leading to an insignificant bud length increase compared to the Co. A total of 24 h after GR, the bud length came to a slight rise and then stabilized. These results indicated that strigolactones can inhibit the activation of dormant buds by decapitation.

### 3.2. Differentially Abundant Protein Analysis

To investigate the differentially abundant proteins (DAPs) associated with bud inhibition by strigolactones, iTRAQ-based proteomic analysis was employed to assess protein changes between the buds on node 2 of the Co, RP, and GR. Functional analysis of DAPs between two compare groups, Co-RP and RP-GR, can provide valuable information for studying the bud inhibition mechanism of strigolactones. A total of 406 DAPs were identified from two compare groups, including 300 DAPs from the Co-RP compare group, 134 DAPs from RP-GR, and 28 DAPs from both (Figure 2A). Among the 300 DAPs from the RP-Co compare group, 253 were up-regulated and 47 down-regulated. A total of 69 DAPs were up-regulated and 65 DAPs down-regulated in the GR-RP compare group (Figure 2B).

### 3.3. Go and KEGG Classification of DAPs Analysis

The most significantly enriched GO categories are displayed in Figure 3, which are classified into three main categories, viz. biological processes (BP), cellular components (CC), and molecular functions (MF). Analysis of BP categories revealed that the DAPs identified in the Co-RP comparison groups were mainly classified into response to heat, S-adenosylmethionine metabolic process, small molecule metabolic process, organonitrogen compound metabolic process, protein import into nucleus, etc. According to MF analysis, the DAPs were classified into nucleotide binding, nucleoside phosphate binding, small molecule binding, unfolded protein binding, etc. According to CC analysis, the DAPs were classified into intracellular, cytoplasm, intracellular part, protein-DNA complex, etc. (Figure 3A).

When comparing the RP and GR groups, the biological process (BP) analysis showed that differentially abundant proteins (DAPs) primarily fall into categories such as precursor metabolite generation and carbohydrate catabolic processes, among others. According to MF analysis: NADP^+^ activity, NAD(P)^+^ activity, NAD binding, oxidoreductase activity, etc. According to CC analysis, GR24 supply mainly affected the intracellular part, photosystem II, nucleosome, DNA packaging complex, etc. (Figure 3B).

To understand the significant responses of cellular processes to activation–inhibition modifications in buds on node two of rice, the KEGG pathway analysis was also performed on the 406 DAPs (Figure 4). According to pathway enrichment analysis, the DAPs in the Co-RP group were significantly enriched in protein processing in the endoplasmic reticulum (Figure 4A). The DAPs in the RP-GR group were enriched in glycolysis/gluconeogenesis, metabolic pathways, and carbon fixation in photosynthetic organisms (Figure 4C). The enrichment analyses based on the proteomics data suggested that glycolysis/gluconeogenesis was most responsive to bud inhibition of strigolactones.

### 3.4. Response of Protein Processing in the Endoplasmic Reticulum and Ribosome Biogenesis to the Effect of RP

Protein processing in the endoplasmic reticulum (ER) and ribosome biogenesis in eukaryotes are two protein synthesis and modification pathways. In our results, these two processes were found most abundant in the RP-Co compare group, specifically (Table 1). A total of 13 out of 19 DAPs involved in the protein processing in ER were upregulated in bud response to RP, which participate in the regulation of protein recognition by luminal chaperones (BiP, GRF94), membrane trafficking (SAR1), ER-associated degradation, and ubiquitin ligase complex. Proteins involved in 90S pre-ribosome components (UTP18) [36], ribosomal RNA (rRNA) modification (NHP2) [37], cleavage (Bms1, KRE33) [38], and other ribosome biogenesis processes (Nug 1/2) were upregulated by RP in the bud. All these results indicate the enhanced protein synthesis in the decapitation-induced bud growth process.

### 3.5. Response of Sucrose Metabolism to the Effect of RP and GR

Glycolysis is an energy-producing process in most living organisms, comprising 10 enzymatic steps from glucose to pyruvate. Further, 12 DAPs were summarized as involved in the glycolysis pathway. Pyruvate kinase was the only protein detected in the buds of the GR-RP compare group, which was strongly up-regulated. The other 11 DAPs were detected in the RP-Co compare group’s buds and were down-regulated (Table 2): three glyceraldehyde-3-phosphate dehydrogenase proteins (GAPC1, GAPC2, GAPC3), a sucrose synthase protein (SUS1), a phosphoglucomutase protein, a fructose-bisphosphate aldolase protein (FBA), a triosephosphate isomerase (TPI), an Enolase (ENO1), a pyruvate kinase (PK15), a pyruvate decarboxylase 2 (PDC2), a malate dehydrogenase (MDH).

To figure out the way of sucrose metabolism and glycolysis are involved in the bud activity and dormancy, we also tested the sucrose content, vacuolar invertase activity, and sucrose synthase activity in buds located at node 2, responding to different treatments (Figure 5). After RP treatment, the sucrose content, vacuolar invertase (vINV) activity, and sucrose synthase (SuSy) activity were strongly enhanced compared to the Co plants (Figure 6). While Extra GR24 supply can somehow reduce the vINV and SuSy activity in buds (Figure 6B,C), it did not affect the sucrose content.

## 4. Discussion

Rice branching is inhibited by the activity of the main panicle (shoot apex). Removal of the panicle can activate the axillary buds to grow. Numerous studies have shown different mechanisms of bud outgrowth after decapitation, and the role of auxin and sugar in apical dormant was the most discussed in the past two decades [8,15,17,19,20,22,39,40,41]. Auxin and sugar represent a highly complex and central signaling network that acts cooperatively in regulating bud outgrowth [18,42]. Glycolysis, TCA cycle, and OPPP (oxidative pentose phosphate pathway) are all the integrative parts of this crosstalk [31]. In addition to auxin, SLs also antagonize sugar stimulation of bud outgrowth. In this study, we investigated what metabolomic processes were involved in the antagonistic effect of sugar and SLs on bud growth.

In the present study, RP and additional GR24 supply act antagonistically in the control of bud outgrowth (Figure 1). In the apical dominance model, sugar, several hormones, and some other components act in a module driving bud release. SLs are phytohormones that inhibit axillary buds, either inhibiting auxin canalization from axillary buds to the main stem or inhibiting bud outgrowth directly [41,43,44,45]. That is why more DAPs were detected in the buds of the RP-Co compare group than in the GR-RP compare group (Figure 2). Through the direct and indirect ways SLs inhibit bud outgrowth, the SLs pathway should be part of the whole regulatory mechanism of decapitation. However, only 28 DAPs were involved in both bud activation and dormant processes (Figure 2A), and most showed the opposite expression (Appendix A). Hence, we hypothesize that the transition between RP-induced bud germination and GR24-induced dormancy buds was carried out through distinct pathways, RP likely triggers more physiological responses than GR, suggesting that the two stimuli activate different mechanisms. The transduction of SL signals is not directly affected by decapitation during the early bud stimulation processes, but it may function downstream of the auxin signal.

Protein processing in the endoplasmic reticulum and ribosome biogenesis in eukaryotes are the two most abundant pathways identified from the KEGG database analysis in the RP-Co compare group but not found in the GR-RP compare group. The endoplasmic reticulum is crucial in protein synthesis, peptide chain folding, and glucose concentration [46,47]. Ribosome biogenesis is a crucial process of protein synthesis and is closely linked to the leading cellular activities, including cell proliferation, differentiation, and growth [48,49]. Based on these results, we suggest that the enhancement of protein processing in ER and ribosome biogenesis might be the quick response to decapitation in axillary buds. Still, SL-induced bud dormant is independent of these processes.

Sucrose is supposed to be an early modulator of other hormones in controlling bud initiation [19,50]. Unsurprisingly, we did not identify DAPs related to auxin biosynthesis, signaling, and metabolism in RP-Co and GR-RP compare groups. Indicating all the DAPs in our results function prior auxin signal. Pyruvate kinase is the key enzyme that catalyzes the final step of the glycolysis pathway, which produces precursors and ATP for various synthetic metabolic pathways [51]. The detected highly abundant pyruvate kinase (PK) in the RP-Co compare group indicates the enhancement of glycolysis after RP. In line with this, sucrose content, vINV and SuSy activities were all enhanced in axillary buds. vINV and SuSy are two enzymes that catalyze sucrose degradation, and they both play an essential role in promoting plant growth [52,53,54,55]. Many of the sugar responses observed in plants are channeled through invertases or sucrose synthases to generate glucose and other signaling sugars to trigger signal transduction via direct perception by diverse sensors or indirect signaling by energy and metabolite sensors [56,57]. Consistent with our results, rapid sucrose accumulation in buds acts in a module to promote bud activation after decapitation, even before auxin depletion in the main stem [58]. That provides more materials for glycolysis. Subsequently, vINV and SuSy activity enhancement indicate more materials for the glycolysis pathway and finally afflux TCA cycle. However, the TCA cycle was not found to be affected after RP, indicating that the TCA cycle was not involved in the bud initiation process caused by decapitation. Decapitation can increase sucrose content, promote its degradation, enhance pyruvate synthesis, and ultimately boost glycolysis in buds.

SLs were regarded as plant hormones that inhibit axillary buds, either inhibiting auxin canalization from axillary buds to the main stem or inhibiting bud outgrowth directly [38,39,40,41]. In our study, sucrose content had not changed in the bud 2 h after GR treatment compared to RP, while sucrose degradation was reduced because of the lower abundance and catalyze activity of key enzymes. It was proved that the sucrose signal can synchronize the formation of lateral roots by promoting the production of auxin in the local area [28,59]. However, when buds are supplied with exogenous SLs, their sugar metabolism processes decrease, which may prevent local auxin biosynthesis. Auxin is necessary for inducing chromatin relaxation and DNA replication [60,61]. This could be the reason why there was still a slight increase in bud length after GR treatment compared to the Co. The increase in bud growth may be due to cytoplasmic growth.

Glycolysis was the most abundant pathway identified from the KEGG database analysis in the GR-RP compare group. Not surprisingly, the TCA cycle was also reduced by SLs. Glycolysis and the TCA cycle provide cell energy that is required for bud initiation [62], and are also considered to be the sugar-signaling-dependent pathways. Our results suggest that the SLs function in buds via the glycolysis-dependent sugar signaling pathways. The reduced sucrose metabolism, including sucrose degradation, glycolysis, and TCA cycle, will not further lower the sucrose content in buds in the short term. Besides the trophic role, sugars can also act as a signaling entity that promotes bud outgrowth [29,63], through glycolysis and TCA cycle [30], which might explain why the sucrose content in buds after GR24 supply remained at the same level as in the decapitation plants. Hence, the bud inhibition function of SLs does not affect sucrose content in buds but directly reduces sugar-signaling-dependent pathways.

## 5. Conclusions

Removal panicle and GR24 supply act antagonistically in the control of buds on node 2 in rice. After decapitation, sucrose is abundant in the bud and many synthetic metabolic pathways are activated before auxin. More importantly, the pathway of sucrose to pyruvate was enhanced for the rest of bud growth. Fewer processes were involved for the SLs’ bud inhibition function, and the sucrose metabolism from degradation to the TCA cycle was impaired without sucrose content changes. This suggests that sucrose metabolism is the key regulatory pathway for the opposing effects of decapitation and SLs on controlling tillers in rice. The inhibition of bud growth by SLs probably relies on the transduction of signals related to sucrose.

## Figures and Tables

**Figure 1 plants-13-00091-f001:**
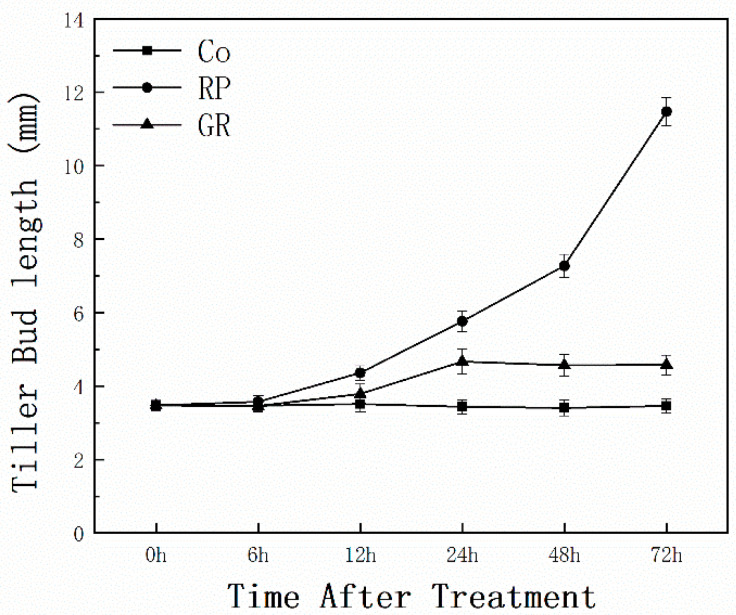
Effects of RP and GR treatments on the length of buds located at the second nodes from the panicle. Rice plants were grown in normal conditions until the full-heading stage and then were subjected to three treatments. Co: intact plants; RP: removed panicle; GR: removed panicle combined with GR24 applied. Vertical bars represent mean ± standard error (*n* = 40).

**Figure 2 plants-13-00091-f002:**
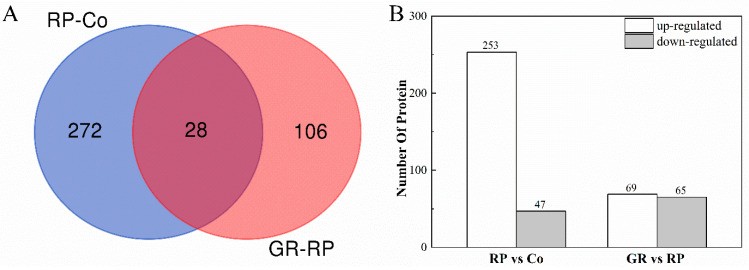
Overview of DAPs in two comparison groups. (**A**) Venn diagram for the DAPs in two comparison groups. (**B**) Numbers of DAPs in each comparison group. The number of DAPs was obtained from comparisons of Co-RP and RP-GR. The DAPs were identified using the restrictive thresholds of fold change ≥ 1.20, *p*-value < 0.05, and unique peptide numbers ≥ 1.

**Figure 3 plants-13-00091-f003:**
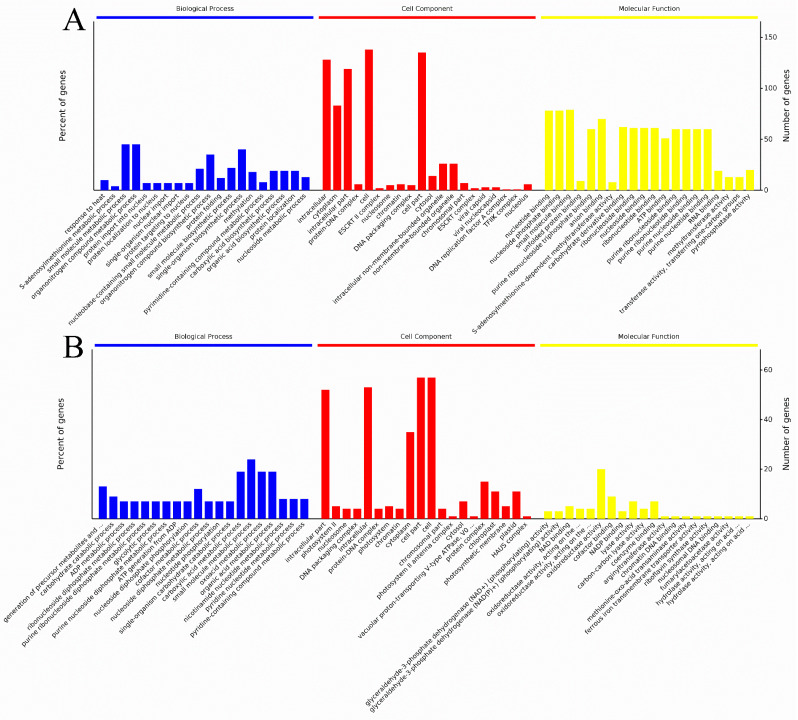
Histogram of significantly abundant proteins in the gene ontology (GO) terms grouped with cellular component, molecular function, and biological process of two comparison groups. Co-RP (**A**), GR-RP (**B**).

**Figure 4 plants-13-00091-f004:**
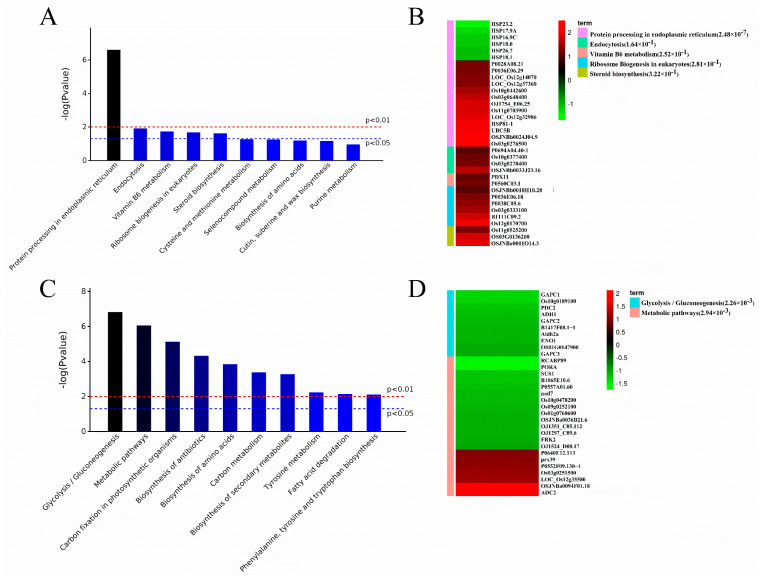
Histogram of significantly abundant proteins in the Kyoto Encyclopedia of Genes and Genomes (KEGG) pathways of two comparison groups. (**A**) Enriched pathways in the buds located at the second node from panicle underlying RP-Co; (**B**) Enriched pathway of DEGs in the buds located at the second node from panicle underlying RP-Co. (**C**) Enriched pathways in the buds located at the second node from panicle underlying GR-RP; (**D**) Enriched pathway of DEGs in the buds located at the second node from panicle underlying GR-RP.

**Figure 5 plants-13-00091-f005:**
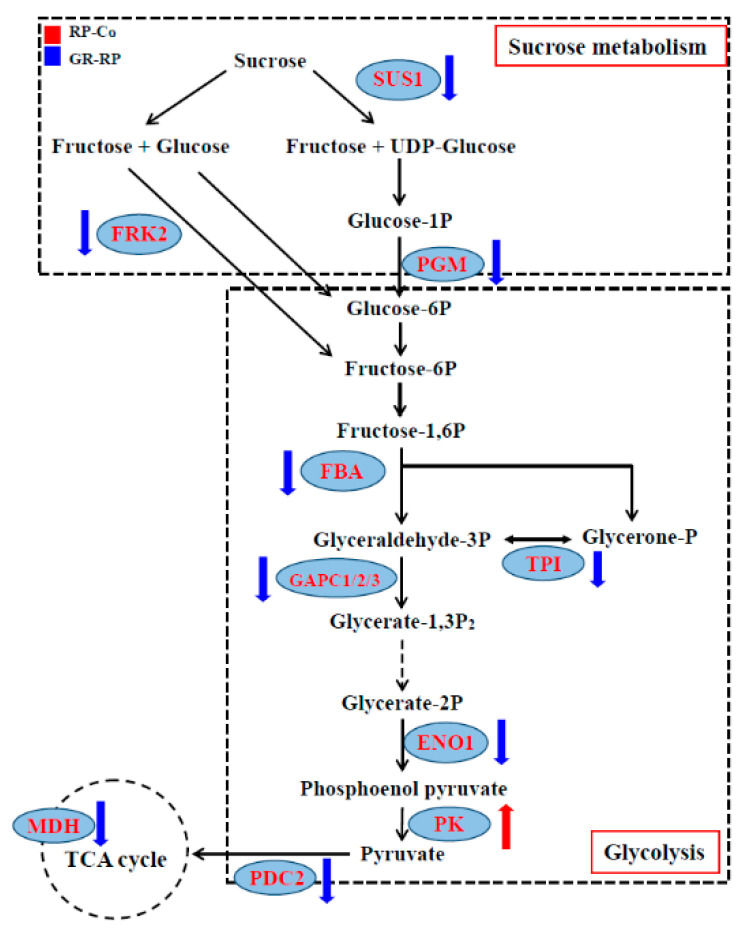
The affected sucrose metabolism processes in the rice bud growth regulation of RP and GR. Up and down arrows are the symbolic representations of increase and decrease, respectively.

**Figure 6 plants-13-00091-f006:**
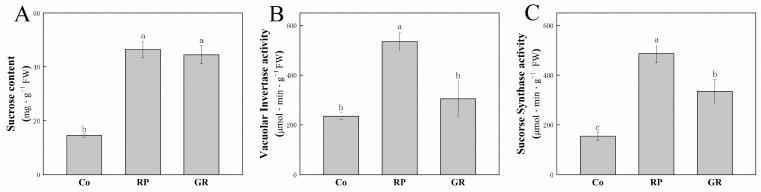
Sucrose content (**A**), vINV (**B**), and SuSy (**C**) activity in the buds. The data represent means from three biological replicates. a, b, c in the picture represents significant difference. Vertical bars represent mean ± standard error (*n* = 40).

**Table 1 plants-13-00091-t001:** DAPs associated with protein processing in endoplasmic reticulum and ribosome biogenesis in eukaryotes in rice bud growth regulation.

Gene Name	ID	RP-Co Fold Change	GR-RP Fold Change	Protein Name
Protein processing in endoplasmic reticulum				
*LOC_Os04g36750*	Q7XUW5	0.3131	NS	23.2 kDa heat shock protein, HSP23.2
*LOC_Os03g15960*	Q84Q77	0.408783	NS	17.9 kDa heat shock protein 1, OsHsp17.9A
*LOC_Os01g04360*	Q943E7	0.44715	NS	16.9 kDa heat shock protein 3, OsHsp16.9C
*LOC_Os01g08860*	Q5VRY1	0.455117	NS	18.0 kDa heat shock protein, OsHsp18.0
*LOC_Os03g14180*	Q10P60	0.4746	NS	26.7 kDa heat shock protein, chloroplastic, OsHsp26.7
*LOC_Os03g16030*	Q84Q72	0.47555	NS	18.1 kDa heat shock protein, OsHsp18.1)
*LOC_Os12g37360*	Q2QNM5	2.52055	NS	GTP-binding protein, SAR1
*LOC_Os02g02410*	Q6Z7B0	2.383533	NS	Luminal-binding protein 3, BIP3
*LOC_Os08g39140*	Q0J4P2	4.257	NS	molecular chaperone HtpG
*LOC_Os03g44620*	Q84PD0	3.266967	NS	DnaJ homolog subfamily A member 2
*LOC_Os03g57340*	Q6F3B0	5.60965	NS	DnaJ homolog subfamily A member 2
*LOC_Os08g38086*	Q6ZCV7	2.38195	NS	heat shock protein 90kDa beta, GFP94
*LOC_Os12g14070*	Q2QV45	2.457367	NS	Stromal 70 kDa heat shock-related protein, chloroplast, putative, expressed
*LOC_Os10g30580*	Q0IXF3	2.8404	NS	cell division control protein 48 homolog E, putative, expressed
*LOC_Os03g60620*	Q84TA1	3.6153	NS	Heat shock cognate 70 kDa protein 2, putative, expressed
*LOC_Os11g47760*	Q53NM9	3.63135	NS	Heat shock cognate 70 kDa protein, putative, expressed
*LOC_Os12g32986*	Q0IN14	3.841017	NS	Hsp90 protein, expressed
*LOC_Os02g16040*	Q8S919	4.47095	NS	Ubiquitin-conjugating enzyme E2 5B
*LOC_Os03g16860*	Q10NA9	5.691567	NS	70 kDa heat shock protein
Ribosome biogenesis in eukaryotes				
*LOC_Os07g41190*	Q6YW01	2.025283	NS	U3 small nucleolar RNA-associated protein 18, UTP18
*LOC_Os02g02360*	Q6Z7B7	2.387583	NS	RNA-binding protein, NOB1
*LOC_Os06g16290*	Q9FP19	2.615183	NS	H/ACA ribonucleoprotein complex subunit 2, NHP2
*LOC_Os03g21530*	Q0DS53	2.894683	NS	ribosome biogenesis protein, BMS1
*LOC_Os01g27730*	Q5ZCV4	3.514833	NS	nuclear GTP-binding protein, Nug2
*LOC_Os12g07300*	Q0IPS3	4.16415	NS	N-acetyltransferase 10, KRE33

NS not significant.

**Table 2 plants-13-00091-t002:** DAPs associated glycolysis in rice bud growth regulation.

Gene Name	ID	RP-Co Fold Change	GR-RP Fold Change	Protein Name
*LOC_Os10g42100*	Q8S7N6	3.188566667	NS	Pyruvate kinase (PK)
*LOC_Os08g03290*	Q0J8A4	NS	0.391	Glyceraldehyde-3-phosphate dehydrogenase 1 (GAPC1), cytosolic
*LOC_Os04g40950*	Q7FAH2	NS	0.455	Glyceraldehyde-3-phosphate dehydrogenase 2 (GAPC1), cytosolic
*LOC_Os02g38920*	Q6K5G8	NS	0.490	Glyceraldehyde-3-phosphate dehydrogenase 3 (GAPC1), cytosolic
*LOC_Os03g18220*	Q10MW3	NS	0.427	Pyruvate decarboxylase 2
*LOC_Os10g08550*	Q42971	NS	0.470	Enolase (ENO1)
*LOC_Os10g11140*	Q33AE4	NS	0.401	Phosphoglucomutase (PGM, alpha-D-glucose-1,6-bisphosphate-dependent)
*LOC_Os01g67860*	Q5N725	NS	0.468	Fructose-bisphosphate aldolase 3 (FBA3), cytoplasmic
*LOC_Os01g05490*	Q0JQP8	NS	0.486	Triosephosphate isomerase (TPI), cytosolic
*LOC_Os03g28330*	P31924	NS	0.425	Sucrose synthase 1 (SUS1)
*LOC_Os08g02120*	Q0J8G4	NS	0.497	Fructokinase-2 (FRK2)
*LOC_Os10g33800*	Q7XDC8	NS	0.451	Malate dehydrogenase (MDH), cytoplasmic

NS not significant.

## Data Availability

Data is contained within the article and Appendix A. The data presented in this study are available on request from the corresponding author.

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
