# Peer review of "Proteomic Analysis Revealed the Antagonistic Effect of Decapitation and Strigolactones on the Tillering Control in Rice"

_plants, 2023, doi:10.3390/plants13010091_

Round 1

Reviewer 1 Report (Previous Reviewer 3)

Comments and Suggestions for Authors

The authors have made commendable efforts to address the major concerns raised in the initial review. They have acknowledged the limitations of their data, expanded discussions on key pathways, and provided additional data for transparency.

Given the progress made and the significance of the study in understanding the role of strigolactones in plant development, I recommend the manuscript for publication in "Plants."

Author Response

Thank you

Reviewer 2 Report (Previous Reviewer 2)

Comments and Suggestions for Authors

This manuscript is improved for new consideration to publish.

A few comments, see file.

Comments on the Quality of English Language

Minor errors should be corrected.

Author Response

we add the soil type. Sandy loam. Line:89

Revised all spelling problems about SuSy. Line: 163,277,278

Revised the spell of trophic. Line: 366

Reviewer 3 Report (New Reviewer)

Comments and Suggestions for Authors

The current study describes the opposite effect of cutting and exogenous strigolactone as transport inhibitors on lateral buds proliferation in rice, Authors have found a large number of differentially expressed proteins during buds outgrowth. Despite a large amount of data, the text lacks physiological mechanism description and shows some fragmentary data on molecular level. The paper can be improved by two ways: either author described detailly physiological mechanism and linked existing data with underlying physiological and epigenetic process including effects of other transport inhibitors. Detailed chromatin analysis of the treatment and effect of exogenous sucrose will be also important to study.

Or authors add information about differently expressed genes, probably some mutants in specific pathways etc.

There are also a number off minor comments which need to be corrected anyway.  

Line 35: „There are two steps for a bud to develop into a tiller: bud initiation and bud outgrowth“ ? Please, read carefully and re-formulate. Bud initiation is not a part of bud development to tiller.

Line 21: „DAPs in germination“? it is better to re-formulate more scientifically.

Line 51: „However, a few phenomena cannot be explained well by these two models[8, 13-15]. In addition, SLs have also been found to regulate auxin transport through the effect of PIN-FORMED1(PIN1) accumulation on xylem parenchyma cell‘ basal membranes[16].

Which phenomena? How PIN1 accumulation regulated auxin transport?

Lines 55- 72: do you mean carbon starvation as building material or carbohydrates (not only sucrose)? https://link.springer.com/article/10.1007/s00425-023-04226-9

Line 73: SLs? exogenous or endogenous??

Line 77: „decapitation and SLs on bud growth“ ?????

Line 82: „Twenty days after seedling,“ ????

Lin 83: „1.6 g urea, 0.8 g 83 plot¿? single superphosphate, and 1.2 g KCl were applied three days before seedling transplantation per pot.¿? How applied¿? Mix or surface?

Surface water was applied to irrigate these pots over the entire growing season of the plants.“ What is surface water?

Lines 88- 96: please, describe all the groups clearly. Here I see only two!

Figure 1: a very good evidence that GR inhibited auxin and sucrose transport and bud possesses only cytoplasmic growth.

Line 198, 207 „biological process“ ¿? I know it is standard, but it seems that all processes related to protein function are biological...

Please, in the discussion part include epigenetic. GR definitely keeps bud dormant by preventing chromatin remodeling through auxin distribution.

English grammar requires some corrections.

Line 275: „The sucrose metabolism is affected in the rice bud growth regulation“ ?? This is not the correct statement.

Line 469: layout.

Comments on the Quality of English Language

Moderate English grammar corrections

Author Response

We carefully reviewed and incorporated all of the comments that you provided. Please see the details below.

Line 35: „There are two steps for a bud to develop into a tiller: bud initiation and bud outgrowth“ ? Please, read carefully and re-formulate. Bud intitaion is not a part of bud develop to tiller.

We had already rewritten this sentence and replaced a new reference. Please see line 35.

Line 21: „DAPs in germination“? it is better to re-formulate more scientific.

It should be “DAPs in germination process”. Please see Line 21-22

Line 51: „However, a few phenomena cannot be explained well by these two models[8, 13-15]. In addition, SLs have also been found to regulate auxin transport through the effect of PIN-FORMED1(PIN1) accumulation on xylem parenchyma cell‘ basal membranes[16].

Which phenomena? How PIN1 accumulation regukated auxin transport?

We have added the details. Line:51-57

Lines 55- 72: do you mean carobon stravation as building material or carbohydrates (not only sucrose)? https://link.springer.com/article/10.1007/s00425-023-04226-9

In this section, we want to emphasize that sucrose could be a crucial factor in the growth of dormant buds into tillers. Sucrose serves as an energy source to facilitate shoot growth, and the metabolites derived from sucrose can act as signaling substances to regulate shoot growth. Thank you for sharing the article. It provided us with a new perspective, and we have cited it. Line 66-68.

Line 73: SLs? exogenous of endogenous??

Here should be endogenous SLs. Line 79

Line 77: „decapitation and SLs on bud growth“ ?????

We corrected the misrepresentation. Line 83

Line 82: „Twenty days after seedling,“ ????

Yes. After 20 days of seeding, the rice plants had 3-4 leaves, which we transplanted for further cultivation in a plastic pot.

Lin 83: „1.6 g urea, 0.8 g 83 plot¿? single superphosphate, and 1.2 g KCl were applied three days before seedling transplantation per pot.¿? How applied¿? Mix or surface?

We applied the nutrition by mixing it with the soil. We have revised this sentence to improve its clarity. Line:m89-91

„Surface water was applied to irrigate these pots over the entire growing season of the plants.“ What is surface water?

It should be rice water layer. We have corrected it. Line: 91

Lines 88- 96: please, decsribe all ther gropus clearly. Here I see only two!

We have only 2 comparison groups in our study, RP-Co and GR-RP. Here we divided all our plants into three groups for three different treatments: control (Co), removed panicle (RP), and GR (2 μM rac-GR24). Line 94-97

Figure 1: a very good evidence that GR inhibited auxin and sucrose transport and bud posses only cytoplasmic growth.

Thank you for sharing your tip. It has been added to the discussion part. Line: 351-357

Line 198, 207 „biological process“ ¿? I know it is standard, but it seems that all process related with protein function is biological...

Can't agree more. GO analysis simply divides the functions of genes into three parts: biological processes (BP), cellular components (CC), and molecular functions (MF).

 Please, in discussion part include epigenetic. GR is definitely keep bud dormant by preventing chromatin remodeliing through auxin distribution.

We have expanded the discussion part. Line: 351-357

English grammar require some corrections.

We have carefully revised the grammar of the full text

Line 275: „The sucrose metabolism is affected in the rice bud grow regulation“ ?? This is not correct statement.

We have corrected it. Line:281

Line 469: layout.

We have corrected it. Line:484

Round 2

Reviewer 3 Report (New Reviewer)

Comments and Suggestions for Authors

Thank you for the corrections. it is lamost good now, Small points still remain. Line 57: citation 16 does not correctly shown that SL inhibiited PIN1 accumulation. Moreover, you need to tell about PIN1 function, not accumulation. Please, read carfully.

Line 88: seedlings or seeding?

Line 205: very interesting point that molecular function is not a biological. Chemical process without link to biology???

Please, correct layout: spaces, comma etc.

Figure 5: what is carbon source in your case? do you have enough active photosyntrehsis in your experiment? Sucrose is a carbon source, but I can nit see this in figure 5.

Comments on the Quality of English Language

Minor layout

Author Response

Thank you for your comments, Thank you for your comments. Once again, we have made modifications and replies according to your latest comments. The details are as follows.

 Line 57: citation 16 does not correctly shown that SL inhibiited PIN1 accumulation. Moreover, you need to tell about PIN1 function, not accumulation. Please, read carfully.

We add the function of PIN1 and replace the citation. Line 55-58.

Line 88: seedlings or seeding?

It should be seeding, we have corrected it. Line 89

Line 205: very interesting point that molecular function is not a biological. Chemical process without link to biology???

Sure, all process related with protein function is biological. We realized that the expression of “BP analysis” is not appropriate, so we re-formulate it. Line 206

Please, correct layout: spaces, comma etc.

We have carefully read through all the manuscript and made some corrections." Line 212, 217, 237, 238, 284, 294.

Figure 5: what is carbon source in your case? do you have enough active photosyntrehsis in your experiment? Sucrose is a carbon source, but I can nit see this in figure 5.

Source is the primary carbon that is transported from the leaf to buds via the phloem. All the rice plants in our research do have enough active photosynthesis. Figure 5 is to display DAPs in our results that are associated with sucrose metabolism.

This manuscript is a resubmission of an earlier submission. The following is a list of the peer review reports and author responses from that submission.

Round 1

Reviewer 1 Report

Comments and Suggestions for Authors

The reason for this decision is:

This manuscript does not fulfill the standards established for the journal to be considered for publication.

Although proteomic analysis revealed the antagonistic effects of decapitation and strigolactone on tillering control of rice, more experiments are required. In addition, the exact function of the gene must be revealed using the product of transformation or genome editing. Otherwise, this paper will cause great confusion to scientists.

Reviewer 2 Report

Comments and Suggestions for Authors

Removing the panicle encourages the growth of buds in rice tillers on the elongated node by getting of apical dominance. Strigolactones (SLs) are plant hormones that suppress tillering in rice. In the present study, the removal of the panicle (RP) and external application of synthesized SLs (GR) was used to regulate rice bud growth on node 2 (the second node from the panicle) at the full heading stage that look inside the proteomic changes in the antagonistic effect on buds: bud germination (RP-Co) and suppression (GR-RP). The method of samplings is not shown and must be introduced. Differentially abundant proteins (DAPs) in germination were most associated with protein processing in endoplasmic reticulum and ribosome biogenesis. DAPs were most associated with metabolic pathways and glycolysis/gluconeogenesis in the bud suppression process. Sucrose content and two enzymes of sucrose degradation in buds were also determined. Comparisons of DAPs between the two reversed processes revealed that sucrose metabolism might be a key to modulating rice bud growth. Moreover, sucrose or its metabolites should be a signal downstream of the SLs signal transduction that modulates rice bud outgrowth. Contemplating the result so far, it is possible to open new vistas of research to reveal the interaction between SLs and sucrose signaling in the control of tillering in rice.This manuscript is in all chapters and paragraph’s well written and gives very interesting new insights in apical dominance of rice and explained well the pathways and metabolism behind these processes.

Comments/remarks, see file.

Reviewer 3 Report

Comments and Suggestions for Authors

The manuscript by Zhao et al. investigates the interplay between decapitation and strigolactone application in regulating rice tillering. Utilizing a proteomics approach, the authors examine specific alterations in protein abundance in rice buds following panicle removal and exogenous GR24 treatment. The authors hypothesize that strigolactones regulate tillering by modulating sucrose metabolism and downstream signaling pathways post-decapitation. Overall, the study is methodologically rigorous and provides an incrementally valuable contribution to the current understanding of the role of strigolactones and sugars in axillary bud development. The proteomics dataset is robust, supported by comprehensive bioinformatic analyses. The manuscript is articulated in a clear, logical manner, and the discussion offers a well-reasoned interpretation of the findings, situating them within extant models. Pending major revisions to address the issues outlined below, the study warrants publication in the journal "Plants."

Major Issues:

-A central assertion of the manuscript is that strigolactones negatively regulate both sucrose metabolism and signaling to limit tillering. However, the available evidence, specifically derived from the assays in Figure 6, does not provide robust support for alterations in sucrose metabolism between RP and GR treatments. Direct quantifications of sucrose, invertase, and sucrose synthase activities show negligible differences between RP and GR buds. Although the proteomics data suggest the involvement of glycolytic enzymes, their functional activities remain unverified. Incorporating additional metabolite profiling could fortify the conclusions regarding sucrose metabolism.

-The proposed model in Figure 5 delineates the conversion of sucrose to pyruvate, yet the evidence indicates that only pyruvate kinase varies between RP and GR conditions. An expanded discussion is warranted to elucidate the full glycolytic pathway, particularly the intermediate steps from sucrose to pyruvate.

-Does the proteomic dataset offer any insights into potential interactions between strigolactones and other phytohormones such as auxin or cytokinin? The manuscript could benefit from a more exhaustive examination of the crosstalk between strigolactones and other signaling molecules.

-The authors postulate that strigolactones do not modulate sucrose levels but specifically impair sucrose signaling mechanisms. This claim necessitates additional clarification. What are the specific components of the sucrose signaling pathway, and how might strigolactones interfere with them independently of altering sucrose concentration?

-To ensure transparency and reproducibility, it is imperative that the authors make available both raw and normalized data for proteomic analysis.

Minor Issues:

-Although the manuscript is generally well-written, there are sections where the clarity and readability of the English could be improved. A meticulous proofreading is advised.

-While the graphical representations are adequately clear, their quality could be augmented by incorporating higher resolution images, embedding fonts, and providing more detailed legends.

Suggested Sentences for Revision:

Abstract, line 15: "The present study employed panicle removal (RP) and external application of synthesized strigolactones (GR) to modulate rice bud growth at node 2. We focused on the full-heading stage to investigate proteomic changes related to bud germination (RP-Co) and suppression (GR-RP)."

Results, line 244: "When comparing the RP and GR groups, the Biological Process (BP) analysis showed that differentially abundant proteins (DAPs) primarily fall into categories such as precursor metabolite generation and carbohydrate catabolic processes, among others. The list of terms would benefit from clarification through rephrasing."

Comments on the Quality of English Language

 The English could be improved in some parts for clarity and readability. I would suggest the authors thoroughly proofread the manuscript.

Here are a few sentences I would suggest revising for clarity and readability:

-Abstract line 15: "In the present study, the removal of the panicle (RP) and external application of synthesized SLs (GR) was used to regulate rice bud growth on node 2 (the second node from the panicle) at the full head-ing stage that we can look inside the proteomic changes in the antagonistic effect on buds: bud germination (RP-Co) and suppression (GR-RP)." This sentence is quite long and difficult to follow. I would suggest breaking it up into 2-3 shorter, clearer sentences.

-Results line 244: "In the comparison of the RP-GR groups, BP analysis revealed that the DAPs were mainly classified into the generation of precursor metabolites and, carbohydrate catabolic process, ADP metabolic process, ribonucleoside diphosphate metabolic process, purine ribonucleoside diphosphate metabolic process, etc."  The list of terms here is confusing due to the punctuation. Suggest rephrasing for clarity.